# IOTA Tangle-based traceability framework for wheat crop supply chain

**Imen Ahmed** [1]*, **Mariem Turki**[1], **Mouna Baklouti**[1], **Bouthaina Dammak**[2]

**1** Computer and Embedded System Laboratory CES, National Engineering School of Sfax, Sfax, Tunisia,
**2** Department of Computer Science, Applied College, Princess Nourah bint Abdulrahman University, Riyadh, Saudi-Arabia

* imen.ahmed@enis.tn

## Abstract

Distributed ledger technology (DLT) has emerged as a transformative solution across industries, offering decentralization, transparency, and security. Among DLT architectures, IOTA's Tangle stands out as a scalable, feeless framework well-suited for supply chain applications, particularly in critical sectors like agriculture. This work leverages IOTA's Tangle to design a secure and traceable wheat supply chain system, addressing challenges such as opacity, fraud, and inefficiency. The proposed solution is an IOTA-based framework designed for enhancing supply chain traceability of wheat, offering trustworthy, transparent, secure and scalable services. Based on smart contracts, a transaction processing system was developed to enhance transparency and traceability and automate tasks. Furthermore, we used various technologies for increased visibility and operational efficiency throughout the supply chain, including IPFS (InterPlanetary File System) for storing extensive data such as stakeholder certifications, AI (Artificial Intelligence) for predicting crop yields, and Self-Sovereign Identity (SSI) for enhanced security. In addition, a government-provided mobile app helps silo managers and police officers verify transportation credentials and prevent contraband. The implementation of this system is expected to enhance supply chain resilience and transparency, particularly in Tunisia, thereby supporting the country's goal of food autonomy.

## 1 Introduction

Food self-sufficiency is one of the major priorities for all independent states. It is even one of the pillars of national security. Every self-respecting independent state ensures the implementation of all necessary strategies, means, and resources to meet the local population's dietary needs and diversity. The objectives are to ensure easy access to all food products for all citizens, stabilize food product prices so they are affordable for all social classes, and combat the monopoly of basic food products. Among these staple products, we can mention milk, sugar, and all grain derivatives

**Data availability statement:** The minimal anonymized dataset necessary to replicate the study findings is publicly available on Kaggle and can be accessed at: https://doi.org/10.34740/kaggle/dsv/14731802.

**Funding:** This work was supported by Princess Nourah bint Abdulrahman University Researchers Supporting Project number (PNURSP2026R847), Princess Nourah bint Abdulrahman University, Riyadh, Saudi Arabia.

**Competing interests:** The authors have declared that no competing interests exist.

such as bread, flour, and semolina. Thus, wheat is considered an essential food element in individuals' lives, particularly in the lives of the Tunisian people. Indeed, Tunisia's culinary heritage is rich in dishes, skills, and products in which wheat is a principal component.

However, in recent years, the entire world has experienced supply difficulties followed by price increases in basic food products such as wheat. Tunisia is not immune to this scourge. In 2021 as well as in 2022, people stood in long lines to buy one or two packages of flour, semolina, or couscous.

According to [1], "Food security is a situation that exists when all people, at all times, have physical, social and economic access to sufficient, safe and nutritious food that meets their dietary needs and food preferences for an active and healthy life." Thus, the last decade has seen remarkable progress in the field of agriculture with the aim of achieving food self-sufficiency. Numerous emerging technologies such as the Internet of Things, Cloud Computing, Blockchain, Big Data, and Artificial Intelligence are being used to implement what we call Agriculture 4.0.

To leverage all these technologies, this work proposes a Wheat Harvest Traceability System, an IoT-based wheat supply chain that includes the Blockchain technology, the Artificial Intelligence, and Self Sovereign Identity to ensure transparency and efficiency. In particular, IOTA was selected as the distributed ledger technology due to its feeless transactions, scalability through its Directed Acyclic Graph (DAG) structure known as the Tangle, lightweight consensus mechanism suitable for IoT devices, and significantly lower energy consumption compared to conventional blockchains such as Ethereum. These characteristics make IOTA especially appropriate for continuous agricultural data exchange and real-time traceability applications.

While this study focuses on the Tunisian wheat supply chain, the proposed IoT and IOTA-based traceability framework can be adapted to other agricultural contexts worldwide, particularly for staple crops where transparency, resilience, and real-time monitoring are critical.

The main contributions of this s tudy are as follows:

1. We proposed a multi-layered architecture for a wheat supply chain, which includes a smart perception layer with an AI module and sensors, a scalable blockchain layer with decentralized storage, and an application layer featuring an interactive web application and a mobile application to verify data integrity.

2. For the blockchain layer, we used IOTA for its efficient and scalable transaction management, enabling secure and tamper-proof data exchange within the supply chain. In our solution, IoT systems located at various locations (vehicles, silos..) can interact with IOTA efficiently, ensuring seamless integration and communication across the entire supply chain. Indeed, to the best of our knowledge, there has been no implementation of a harvest supply chain in agriculture using IOTA smart contracts.

3. Enhancing System Security with Self Sovereign Identity: In this system, the farmer acts as the issuer, the transporter as the holder, and the silo responsible as the verifier. This SSI approach ensures secure and verifiable identities

and transactions throughout the supply chain, effectively preventing unauthorized access and ensuring high network security.

4. We have implemented the entire system and evaluated the performance of the IOTA network in terms of energy consumption, cost, and throughput, comparing it to the Ethereum blockchain.

The remainder of the paper is organized as follows: Section 1.1 reviews existing studies on agrifood supply chains. Section 2 provides an overview of the proposed wheat supply chain, detailing interactive component layers and the integration of advanced technologies like blockchain, IoT, and SSI. Section 3 describes the system implementation and tools used. Section 4 analyzes the system's outcomes, evaluating performance, efficiency, and impact. Finally, Section 5 concludes the paper.

## 1.1 State of the art

Various state-of-the-art solutions have been proposed for smart agriculture based on IoT, such the works described in [2] and [3]. These solutions allow real-time collection of data related to crops and livestock, significantly enhancing productivity and product quality while simultaneously reducing costs. However, these solutions are based on centralized architectures, which poses significant security risks. Recently, advancements in blockchain technology have significantly enhanced the traceability and transparency of agricultural products. Blockchain's decentralized ledger ensures immutability and validity of transactions, which is crucial for maintaining a reliable traceability system in the agriculture sector. The integration of blockchain with the InterPlanetary File System (IPFS) has addressed the storage limitations inherent in blockchain by allowing off-chain storage of large datasets while ensuring data integrity and accessibility through cryptographic hashes stored on the blockchain [4].

Blokchain has been widely used to guarantee agricultural traceability, as reported by different reviews and industry reports. As an example, the UNDP (United Nations Development Programme) report highlighted the role of blockchain in implementing agri-food supply chains [5]. [6] explored the role of blockchain technology in modern high-value food supply chains. It has been noted that blockchain is mainly used to assure food safety and prevent frauds. The works presented in [7] and [8] proposed several blockchain-based solutions for food product traceability. However, only theoretical concepts are presented without any experimental implementation or performance evaluation. A review of supply chain management systems, addressing the use of blockchain technology, was also provided in [9]. Zhang et al., proposed a new architecture based on blockchain for the grain supply chain to address safety-food issues [10]. The authors designed a multimode storage mechanism that combines chain storage. A prototype system was instantiated based on Hyperledger Fabric and tested among different scenarios. However, the proposed system still has some deficiencies in supply chain information management. Similarly, another system for blockchain-based agriculture and food supply chain, named Agri-Food, was described in [11]. Shahid et al., used smart contracts to manage transactions and the IPFS storage. Although simulations have been carried out and evaluated, issues like scalability, interoperability, immutability and efficiency are not covered in this study. Yakubu et al., [12] proposed a blockchain-based framework to assure reliability and efficiency while monitoring and tracking rice products safety and quality, particularly during product purchasing. The system has been evaluated in terms of performance and cost. However, this work has not addressed electronic payments and proof of delivery, essential to guarantee the validity and security of commercial transactions. In [13], Gondal et al., developed a blockchain-based strawberry supply chain framework to track the movement of strawberries from the farm to the consumer in a secure and a transparent way. They used Ethereum smart contracts to manage transactions between different actors. They also monitored the status of IoT containers, mainly to control temperature, humidity, and location of shipment, for food supply chains and notify consumers. The framework has been only tested via simulation and no real deployment has been made. In addition, no dApp has been developed for this supply chain. In [14], Farooq et al., proposed a blockchain-based framework for the wheat crop supply chain traceability. They used a crypto token with Ethereum-based smart contracts to

keep track of transactions among the different involved actors. The implemented system has shown better performance compared to both Bitcoin and Ethereum. Meanwhile, as the system is oriented to financial transactions, a deep security analysis needs to be carried out. These aforementioned implementations demonstrate blockchain's ability to provide an immutable record of each transaction, thereby improving consumer trust and supply chain efficiency.

## 2 Wheat supply chain description

In this section, we showcase our suggested method for attaining traceability along the wheat supply chain, guaranteeing clarity, effectiveness, and safety. The interactions between the various components of the system are depicted in Fig 1 This architecture involves several actors, each with specific responsibilities. The farmer is responsible for growing and harvesting the wheat. Once harvested, the farmer selects a transporter and the destination silos based on ratings, weather conditions, location, capacity, etc. The transporter is then responsible for delivering the wheat to the designated silo. The silo serves as the storage location for the wheat. Additionally, a verifier, who could be a policeman, is tasked with verifying the credentials of the transporter to ensure the authenticity and security of the transportation process.

### 2.1 Smart perception layer

In the proposed solution, we introduced two IoT systems: one located in the transporter's vehicle and the other in the silo.

- Vehicle IoT System: The vehicle subsystem includes a weight sensor, a GPS sensor, and temperature and humidity sensors. Throughout transportation, these sensors collect information on the load and the surrounding conditions.

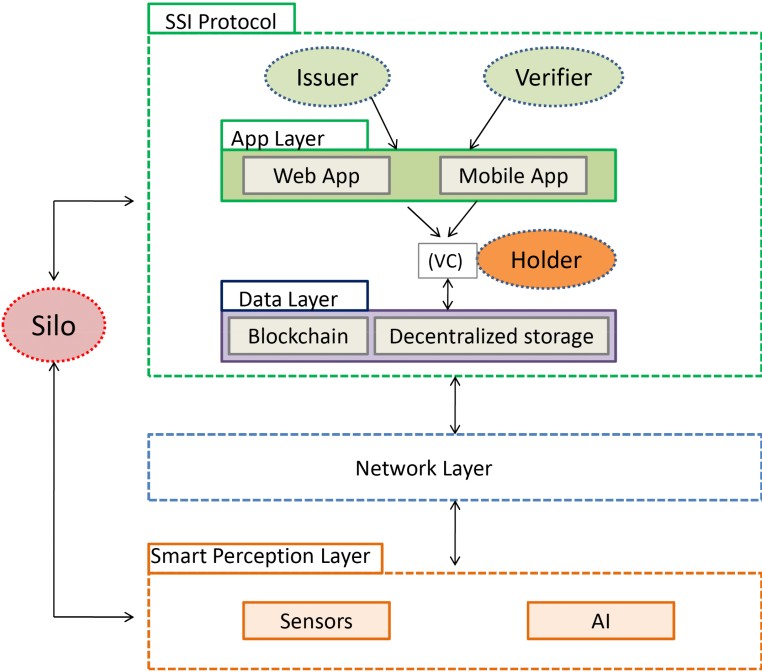

**Fig 1. General Architecture Overview.** The SSI protocol involves Issuer, Holder, and Verifier roles interacting through web and mobile applications. Verifiable Credentials (VCs) are exchanged and managed over a data layer combining blockchain and decentralized storage. The network layer enables communication with the smart perception layer, which includes sensors and AI modules. An external silo represents an isolated data source interconnected with the system.

- Silo IoT System: it also includes weight, humidity, and temperature sensors, as well as a camera for capturing images of the wheat. An artificial intelligence model evaluates these photos to assess whether the wheat is healthy or damaged. This integration of AI for real-time image analysis with IoT sensors for data collection is why the system is called the Artificial Intelligence of Things (AIoT) System, combining the capabilities of both AI and IoT to enhance efficiency and reliability in monitoring and managing the wheat supply chain. The AIoT system is based on a basic Convolutional Neural Network (CNN) model integrated into an embedded platform. The dataset includes 1188 images of wheat, with 706 depicting good wheat and 482 depicting damaged wheat. Data augmentation techniques such as rotations and flips were applied to increase variability. The CNN consists of multiple convolutional layers for feature extraction followed by fully connected layers for classification. However, the details of this architecture and model optimization are not covered in this paper, as the AI component is not the main focus of this study.

## 2.2  Network layer

The smart perception layer and the upper ones communicate with each other through the network layer.

In our paper, we have chosen to use WiFi communication due to its higher data transfer rates, lower latency, and widespread availability, which are crucial for our AIoT system that requires real-time data processing and high-volume data transmission. Data will then be transported via routers and gateways to ensure seamless connectivity and efficient data flow throughout the system.

## 2.3  Data layer

The Data Layer is a critical component of our system, integrating blockchain technology and decentralized storage to ensure data integrity, security, and accessibility. Together, these technologies form a robust Data Management Layer that underpins the entire system, facilitating secure data sharing and collaboration across the network.

**2.3.1  IOTA-based blockchain platform.**  We leverage IOTA Distributed Ledger Technology (DLT) to establish the proposed Wheat Harvest Traceability supply chain and the smart contract designed to monitor and facilitate transactions within this supply chain.

In our exploration of current technologies in the agricultural field depicted in Table 1, we have observed that the majority of proposed blockchain systems integrating smart contracts are based on platforms like Ethereum and its extended frameworks (such as Quorum), or the Hyperledger Fabric platform. In these traditional blockchain systems, transactions are organized into blocks, each containing a limited number of transactions bundled together in a linear chain. In contrast, IOTA diverges from this model significantly by employing a Directed Acyclic Graph (DAG) structure known as the Tangle, which allows for a more decentralized and scalable approach to transaction validation [15].

**Table 1.  Blockchain-based solutions in the agricultural field.**

| Similar work | Blockchain | Smart contract | IPFS | Year |
|---|---|---|---|---|
| [17] | Ethereum | Yes | No | 2024 |
| [14,18] | Ethereum | Yes | Yes | 2024 |
| [19] | Ethereum and Quorum | Yes | No | 2024 |
| [20] | Ethereum | Yes | Yes | 2023 |
| [21] | Ethereum and Hyperledger | Yes | Yes | 2022 |
| [22] | Hyperledger | Yes | No | 2022 |
| [10] | Hyperledger | Yes | No | 2020 |
| [23] | IOTA | No | No | 2019 |

In the Tangle, each new transaction must approve two previous transactions, effectively creating a network where transactions are interlinked rather than linearly chained. Hence IOTA's Tangle theoretically becomes faster and more efficient as more transactions are added, since each new transaction contributes to the validation of previous transactions [16].

**2.3.2 IOTA tangle suitability for IoT.** The IOTA Tangle has gained significant attention in the literature for its suitability in addressing the challenges of IoT [24,25]. Unlike traditional blockchain technologies, the Tangle offers a flexible structure that can be adapted to meet the diverse needs of IoT systems. IoT currently faces several limitations, such as scalability, high transaction costs, and security vulnerabilities, all of which can be effectively addressed through the unique features of the IOTA Tangle. Below are the key benefits and reasons for integrating IOTA technology into IoT infrastructure:

- Scalability: As IoT continues to expand, a scalable infrastructure is essential to handle the increasing volume of devices and data. The IOTA Tangle's architecture is inherently scalable, allowing the network to grow seamlessly with the addition of new participants [25].

- Security: In IoT systems, ensuring data integrity and protecting against cyberattacks are critical concerns. The IOTA Tangle enhances security through its decentralized structure. Each transaction must approve two previous ones, creating a web of interlinked transactions that makes it extremely difficult for malicious actors to alter or manipulate data [24].

- With many IoT devices operating on limited power sources, energy efficiency is crucial.The IOTA Tangle is designed to be lightweight and energy-efficient, as it doesn't rely on energy-intensive mining like traditional blockchain networks [25].

- Transaction Efficiency: The lightweight and feeless nature of IOTA transactions make it especially suitable for IoT applications. In environments where microtransactions and high transaction volumes are the norm, IOTA's design ensures that these operations can be carried out seamlessly without incurring fees or requiring significant computational resources [26].

**2.3.3 Tip selection algorithm on IOTA Tangle.** As we have mentioned above, instead of relying on miners to validate transactions, in IOTA each transaction must select and confirm two previous transactions, a process known as tip selection. In the IOTA Tangle,as depicted in Fig 2, unconfirmed transactions at the edges of the network graph are referred to as "tips."

To confirm a new transaction, a user must choose two tips to validate, which involves a tip selection algorithm. This algorithm plays a crucial role in maintaining the efficiency, security, and speed of the network. Several types of tip selection algorithms have been proposed, ranging from simple random selection such as Uniform Random Tip Selection(URTS), to more complex approaches like weighted random walk(WRW).

Indeed, in our previous work presented at the international conference (AICCSA) [27], we introduced an optimized tip selection algorithm named Age Aware-Weighted Random Walk (AA-WRW), designed to enhance the tip selection process. AA-WRW improves upon WRW by reducing the number of permanent tips and increasing selection efficiency, contributing to a more robust and scalable Tangle structure. After analyzing the Weighted Random Walk (WRW) algorithm, we identified its limitation in maintaining a high number of permanent tips. A prmenant tip is a Transaction that validates other transactions but never get validated by any subsequent transactions, making it a "stuck" or "unconfirmed" transaction in the Tangle.

This limitation affects network performance by leading to inefficiencies in transaction validation, as an excessive number of permanent tips can cause delays in transaction finality. Additionally, the accumulation of permanent tips increases the computational overhead, which reduces the overall throughput and scalability of the network, ultimately hindering its ability to handle large volumes of IoT data efficiently. To address this, we proposed and developed a new algorithm called AA-WRW, which aims to improve tip selection by rewarding older tips and ensuring a more efficient network. The AA-WRW algorithm builds on the WRW by first filtering the most well-supported tips, then introducing an additional

 

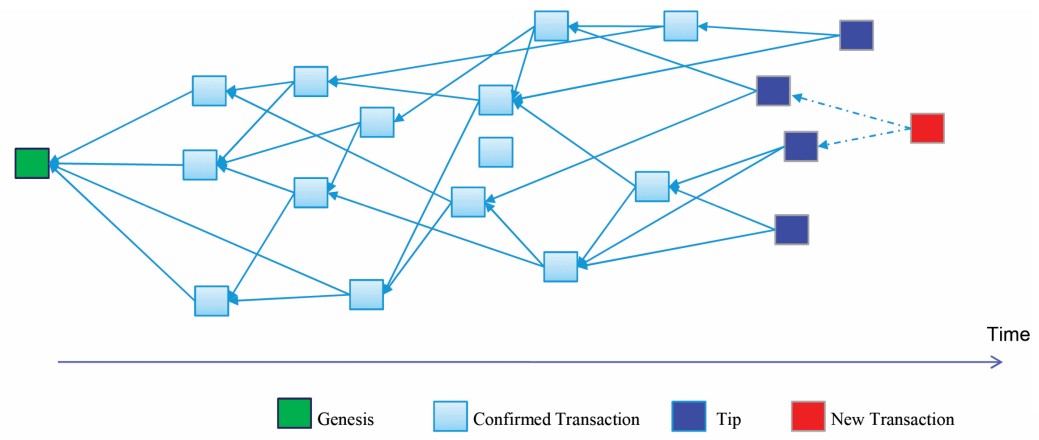

**Fig 2. Tip selection on IOTA Tangle.** The figure illustrates a DAG-based ledger where each new transaction approves two previous tips. Confirmed transactions propagate from the genesis node, while tips represent unapproved transactions. The red node indicates a newly issued transaction awaiting confirmation over time.

criterion that prioritizes tips based on their age. This approach ensures that more stable and reliable tips are chosen. This two-step approach reduces the number of permanent tips while improving the robustness of the selections and enhancing the overall efficiency of the network. Experimental results demonstrate the effectiveness of our optimization, achieving a reduction in permanent tips by over 30% compared to the original WRW approach.

**2.3.4 IOTA in our proposed Harvest supply chain.** IOTA has a variety of roles in this proposed framework. Firstly, it facilitates the creation of a tamper-proof record of each step in the wheat supply chain. Every transaction, is cryptographically secured and timestamped on the Tangle. By enabling stakeholders to confirm the legitimacy and provenance of items, this transparency boosts confidence and lowers the possibility of fraud or counterfeiting [24]. Secondly, IOTA enables real-time data collection and sharing among relevant parties. The suggested IOTA-based platform communicates with the IoT sensors to offer secure records of numerous transactions and transportation-related incidents pertaining to IoT environmental measures.

Furthermore, smart contracts are considered a significant factor in this model, which automate many tasks. Thereby, by authenticating their cryptographic keys, smart contracts allow only authorized actors to connect or access to the system.

## 2.4 Application layer

The application layer in the wheat supply chain includes three main components: the SSI (Self-Sovereign Identity) protocol, the web application, and the mobile application. These components are tightly integrated and interact closely with each other.

**2.4.1 SSI protocol.** Self-sovereign identity (SSI) is a digital identity management model that allows individuals to control their personal information independently using decentralized technologies such as blockchain. In the context of our wheat supply chain, SSI is particularly useful for managing the identities and credentials of different actors involved in the process. The farmer, who grows and harvests the wheat, uses SSI to issue credentials to the transporter. These credentials, which include relevant details about the transportation task, are tied to the farmer's Decentralized Identifier (DID). This system ensures that the transporter's credentials are securely issued and can be verified by authorized parties. The verifier, such as a policeman, can then use these credentials to confirm the transporter's legitimacy and

compliance with regulations. The use of Verifiable Credentials (VCs) ensures that the credentials are secure and tamper-proof, thanks to cryptographic elements that verify the integrity of the data. This approach enhances privacy, security, and trust throughout the wheat supply chain, allowing each party to verify credentials without relying on intermediaries.

**2.4.2 Web application.** In this part, we highlight some of the main features of the web application that are offered to the various involved actors.

- All users, including Farmers, transporters, and Silo Responsibles, have access to a profile section where they can view their information and update their passwords to enhance security.

- Farmers can request transportation services from transporters, select their produce, and schedule the transfer of the harvest by choosing a suitable date based on weather conditions. They also determine the destination based on the silo's location and availability. Additionally, farmers have the option to choose a transporter based on their rating. Upon selection, they provide the transporter with authenticated credentials to ensure proper delivery verification to both law enforcement and the receiving silo.

- To effectively oversee wheat storage and monitor inventory levels, Silo Responsibles can gain insight into current stock levels. Additionally, they have the capability to track incoming transportation, ensuring they are promptly informed of deliveries. They can then confirm receipt of the transported goods.

- At the end of the transportation, the farmer can rate each transport service based on several factors, including the speed of the service and adherence to transportation conditions such as temperature and humidity.

**2.4.3 Mobile application.** The mobile application allows police officers and verifiers to capture and scan QR codes associated with the farmer's credentials. Using the app, one may quickly verify credentials using the blockchain-based Self-Sovereign Identity (SSI) system by scanning a QR code. Subsequently, the mobile application shows a notification verifying the validity of the credential, guaranteeing prompt and safe document authentication for the farmer.

## 3 Implementation of proposed Harvest supply chain

In this section, we present the implementation environment of the whole system. All the tools and frameworks are illustrated in Fig 3.

### 3.1 IoT system and sensor networks

In this section, we detail the hardware components of our IoT system.

The first IoT system is located in the transporter's vehicle. This system includes a DHT11 sensor used to monitor the temperature and humidity levels inside the vehicle, ensuring the wheat is transported under optimal conditions. Additionally, the setup features an HX711 load cell with a 20 kg capacity to measure the weight of the wheat during transit, ensuring that the load remains consistent and within expected parameters. The DHT11 and HX711 sensors were calibrated according to the manufacturer's instructions prior to each data collection session. Data were recorded every 15 minutes to continuously track temperature, humidity, and weight. To provide real-time tracking of the vehicle's location, a Neo-7M GPS module is integrated into the system. This GPS data enables the tracking of the wheat's journey from the transporter to the retailer or storage facility, helping to monitor the route and ensure timely deliveries The Neo-7M GPS module recorded the vehicle's location every 30 seconds, providing real-time tracking throughout the transport process. The second IoT system is installed in the silo. Similar to the vehicle setup, this system uses a DHT11 sensor to monitor the temperature and humidity within the silo, maintaining optimal storage conditions to preserve the quality of the wheat. An HX711 load cell with a 20 kg capacity is also employed here to measure the weight of the wheat as it is stored or removed from the silo, aiding in inventory management and monitoring the quantity of wheat in storage. Additionally, a Pi Camera is included to capture images of the wheat in the silo. These images, captured hourly, are used for visual inspections to

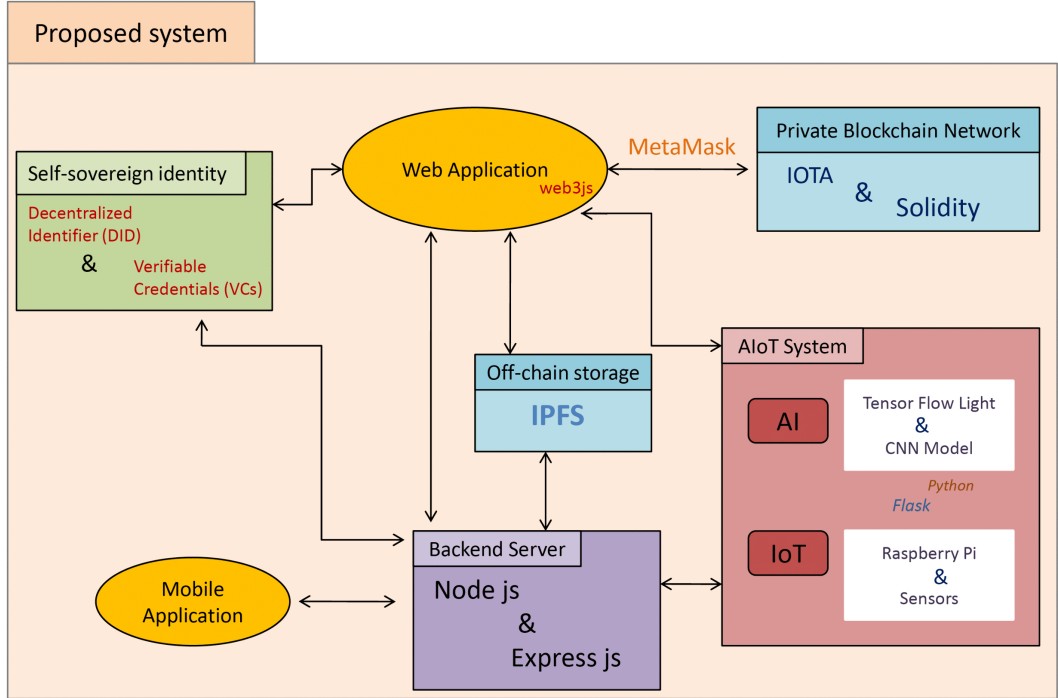

**Fig 3. The Implementation environment of the Wheat Supply Chain.** The system integrates a Self-Sovereign Identity framework, web and mobile applications, a private IOTA-based blockchain network, off-chain storage using IPFS, and an AIoT layer combining AI modules and IoT sensors. A back-end server coordinates interactions across these components to support secure data exchange within the supply chain.

identify potential issues such as mold or pest infestations, ensuring that the wheat remains in good condition. Together, these IoT systems provide a comprehensive monitoring solution that ensures the wheat is kept in optimal conditions throughout its journey in the supply chain. The integration of these sensors allows for real-time data collection and monitoring, facilitating better decision-making and enhancing the overall efficiency and reliability of the wheat supply chain.

## 3.2 Blockchain implementation

In this section, we outline the main functions of the smart contracts written in solidity and deployed on both Ethereum and IOTA blockchain platforms to compare the performances of these two blockchains.

### 3.2.1 Smart contract implementation and cybersecurity measures.
We developed one smart contract, which includes every aspect required to handle participants while tracking food back to its origin using the blockchain technology. The full source code of the developed smart contract is available in [28]. The smart contract has various data structures and functions for dealing with the roles of various actors in the supply chain such as farmers, transporters and silos.

In order to evaluate our smart contract, we conducted a security analysis using the Mythril tool [29]. Mythril is a symbolic execution tool equipped with built-in detection modules designed to identify bugs such as integer overflows and reentrancy vulnerabilities. Fig 4 shows that the tool has not reported any issues or vulnerabilities.

In addition to the Mythril static analysis results, several common attack vectors for Ethereum smart contracts, such as reentrancy, integer overflow, and unauthorized access, were considered during development. These were mitigated by:

- Restricting access to critical functions through role-based modifiers (onlyOwner, onlyFarmer, onlyTransporter, onlySilo, and onlyAuthorized).

```
mythril@9da1c08e6fb2: ~          ×     +   ∨
mythril@9da1c08e6fb2:~$ myth analyze wheat.sol
The analysis was completed successfully. No issues were detected.

mythril@9da1c08e6fb2:~$
```

**Fig 4. Analysis results with Mythril tool.** The figure presents the results obtained from analyzing the smart contract for potential vulnerabilities and security issues with the Mythril tool.

• Avoiding external contract calls to eliminate reentrancy risks.

• Relying on Solidity version ≥ 0.8.0, which provides built-in overflow and underflow protection.

Additionally, sensitive user information is represented through decentralized identifiers (DIDs) rather than raw credentials, ensuring privacy preservation and improving cybersecurity by securing data transactions and preserving users' anonymity [30]. These design decisions collectively enhance the robustness and integrity of the proposed traceability contract.

**3.2.2 Blockchain configuration.** We have developed a harvest supply chain system that leverages the core advantages of the IOTA Tangle and its smart contract protocol (ISCP) framework.

IOTA consists of two types of nodes: Hornet nodes, which implement Layer 1, and Wasp nodes, which implement Layer 2 [31]. Hornet nodes play a pivotal role in the IOTA network by verifying transactions, handling client requests, managing the Tangle's state, and executing necessary protocols to maintain network functionality. However, Wasp nodes are specifically designed to enhance the execution of smart contracts and decentralized applications (dApps) on the IOTA network [32]. While these nodes are typically distributed across multiple computers to ensure ledger consensus, our current implementation confines each to a single instance within the local network for testing purposes. While this single-instance setup was sufficient for validating the system's functionality and end-to-end data flow, in a real-world deployment, multiple distributed Hornet and Wasp nodes would be required to evaluate network resilience, latency, and consensus performance under higher transaction loads (Fig 5).

After establishing a chain within the Wasp node, it generates a JSON-RPC that points to the deployed chain as shown in Fig 6. This JSON-RPC is essential for interacting with the chain effectively. We proceeded to implement the smart contract using hardhat framework [33], EVM ChainID and JSON-RPC URL.

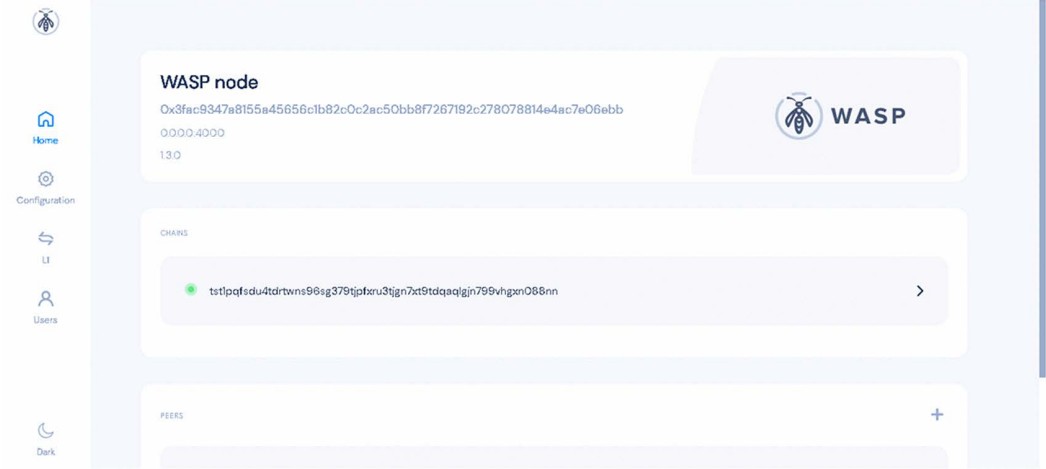

**Fig 5. Wasp node dashboard.** The figure illustrates the dashboard of the Wasp node implemented on a Raspberry Pi board.

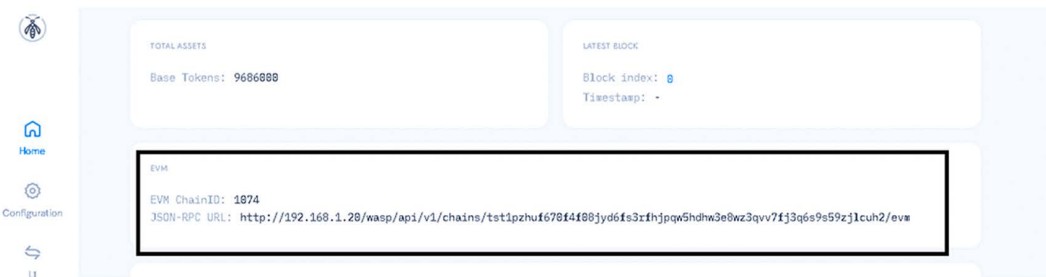

**Fig 6. Chain Id and JSON RPC URL.** The dashboard displays the Chain ID and the JSON-RPC URL, which are essential parameters for interacting with the blockchain network.

To evaluate the effectiveness of the IOTA-based harvest supply chain, our approach involves implementing the smart contract on the IOTA network first, followed by Ethereum, and comparing the outcomes of both implementations.

Both IOTA and Ethereum are blockchain networks compatible with the Ethereum Virtual Machine (EVM). To conduct thorough testing, we have developed a local private network on a Raspberry Pi board. Initially, we set up the IOTA network comprising Hornet and Wasp nodes, followed by an Ethereum network consisting of two nodes, to make the result comparable, created using the Geth client.

### 3.3 SSI implementation

We implement Self-Sovereign Identity (SSI) using the Veramo framework [34], which is written in TypeScript language. Veramo provides a robust infrastructure for managing decentralized identifiers (DIDs) and verifiable credentials, essential for ensuring the security and integrity of identity data within the harvest traceability supply chain. We selected Veramo over other SSI implementations due to its modular architecture, interoperability with W3C DID(World Wide Web Consortium – Decentralized Identifier) standards, and lightweight integration capabilities, which make it particularly suitable for IoT-based environments requiring efficient identity verification and communication between distributed nodes. Veramo offers high flexibility for lightweight IoT devices and provides faster DID resolution, which are critical requirements in real-time agricultural supply chain scenarios. The Veramo environment is set up in *setup.ts*, initializing plugins and services for DID management and verifiable credentials. *createIdentifier.ts* generates new DIDs on the Sepolia test network for supply chain stakeholders. *createCredential.ts* issues verifiable credentials, such as farmerName, transporterName, and weight. *verifyCredential.ts* verifies the authenticity and validity of these credentials, ensuring only trusted ones are used in the system.

To facilitate interaction with our SSI system, we use Express.js to create a RESTful API. This interface allows external applications and users to interact with the Veramo-based SSI system, enabling operations such as DID creation, credential issuance, and credential verification.

All issued credentials are stored on an IPFS node.

For credential verification, we use a React Native-based mobile app. The app scans a QR code that contains the hash of the credential stored in IPFS. Upon scanning, the app retrieves the credential from IPFS and verifies it using the *verifyCredential.ts* script. This process ensures that the credentials presented are authentic and have not been tampered with. Fig 7 illustrates some interfaces of the mobile application.

The figure in the middle presents the results of scanning a QR code with authentic credentials, while the figure on the right illustrates the scan of a QR code with incorrect credentials.

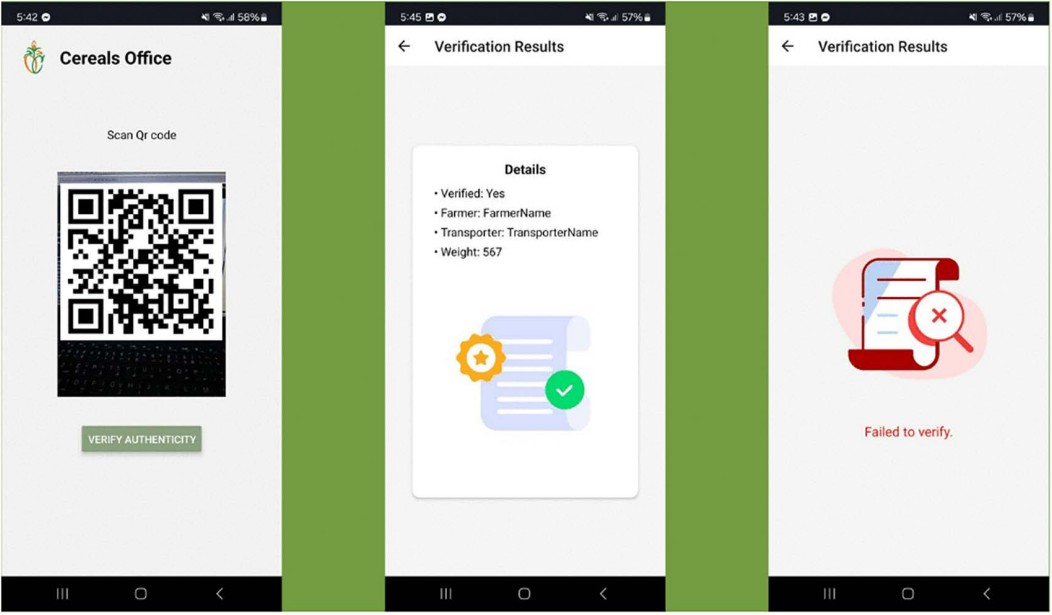

**Fig 7. Mobile application for credentials verification.** The figure shows the mobile app interface, including the authentication page and the verification page displaying authenticity checks and verification results.

## 3.4 Web application

In our project, we developed the web application using ReactJS and React Bootstrap to create a responsive and interactive user interface. Real-time sensor data from the IoT system is transmitted via WebSocket (Socket.IO), enabling to track the transportation progress in real time. For the GPS sensor, the interface employs the React Router to calculate and display the best route based on the current location and dynamically changing GPS data, ensuring efficient navigation and timely delivery of the yield. The interactive map is implemented using the Leaflet library. In addition, we have used OpenWeather to implement the weather station functionality in the web application. Fig 8 shows an example of the web interface of the silo.

## 4 Evaluation and performance analysis

To assess the effectiveness of the IOTA-based harvest supply chain, our methodology includes initial implementation of the smart contract on the IOTA network, followed by Ethereum, with subsequent comparison of their respective outcomes, as we mentioned. This evaluation encompasses metrics including energy consumption, cost and throughput across both platforms.

### 4.1 Energy consumption

Our experiment involved measuring the energy consumption of transactions on both the IOTA and Ethereum networks. We started by deploying smart contracts and subsequently executed various functions, closely monitoring the power usage of the Raspberry Pi for each operation. Each transaction was repeated ten times to ensure precise measurements, with an average deviation of approximately two minutes between executions. The power consumption in watts was measured using USB voltage- and current-detection module. The Raspberry Pi operates on a power supply of 5 volts with a current rating of 2.5 amps, and it typically consumes around 500 milliamps on average.

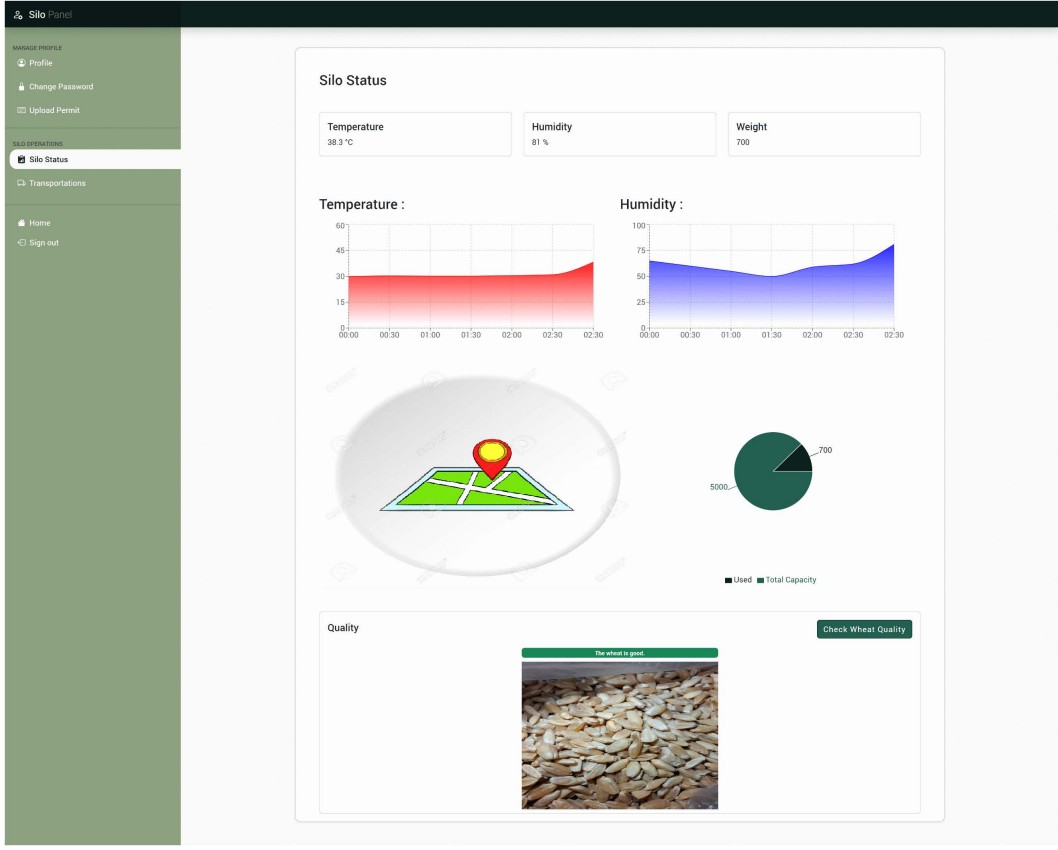

**Fig 8. The Silo Status Interface.** The interface displays the current status of the silo, including temperature, humidity, and weight. Users can check wheat quality using the provided button. The figure includes graphs showing temperature and humidity trends over time, a chart indicating used and total wheat quality, and a map indicating the location of the silo.

The power measured in watts is calculated using Equation (1).

$$W = A \times V \qquad (1)$$

where W represents power, A represents current intensity, and V represents voltage (which is 5 volts).

We observed and recorded the power consumption for transactions executed on both Ethereum (using proof-of-work consensus) and IOTA (utilizing the Tangle). The results, as depicted in Fig 9, show a consistent pattern where IOTA transactions consume approximately 13% less energy compared to Ethereum transactions. This difference can be attributed to the energy-efficient design of IOTA's Tangle consensus mechanism, which does not rely on intensive computational mining processes like Ethereum's proof-of-work. In practical terms, this reduction is highly relevant for large-scale agricultural operations and IoT-based supply chains, where thousands of transactions may occur daily. Lower energy consumption reduces operational costs and enables the deployment of lightweight IoT nodes powered by limited energy sources (e.g., solar or battery). Moreover, it contributes to the sustainability goals of green agriculture by minimizing the environmental footprint of digital traceability systems. This finding underscores the potential of IOTA's Tangle to significantly reduce energy consumption in decentralized applications, offering a promising alternative for sustainable blockchain solutions.

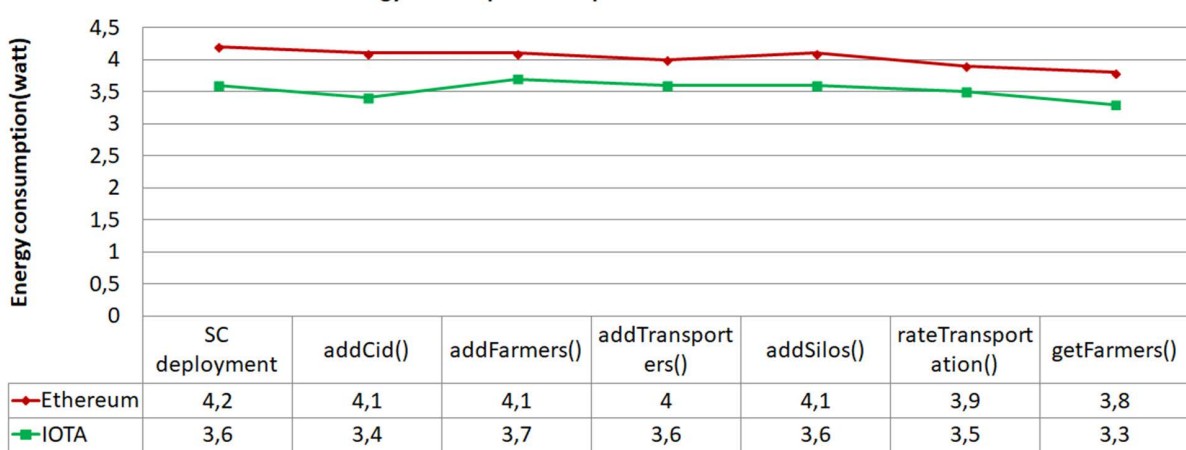

**Fig 9. Energy consumption Comparison: Ethereum vs IOTA.** The figure compares the energy consumption of the two blockchains for different transactions, including smart contract deployment and the execution of individual smart contract functions.

## 4.2 Transactions cost

We conducted a comparison of the costs associated with various functions of the smart contract executed on both the Ethereum blockchain and the IOTA network. As illustrated in the graph in Fig 10, transactions on the IOTA network are significantly more cost-effective compared to Ethereum, being almost negligible in cost. For instance, executing addTransportation() costs 0.00081361 Ether on Ethereum, whereas it costs only 0.00308984 IOTA, equivalent to approximately 0.000000153 Ether based on the exchange rate of 1 IOTA = 0.00005 ETH on July 17, 2024. The deployment of the contract was also cheaper, costing only 0.00000028 ETH compared to Ethereum's 0.019 ETH.

## 4.3 Throughput results

Throughput refers to the number of transactions processed per second by a blockchain network. It is a critical metric indicating the network's capacity to handle transactions efficiently. To measure the throughput and block confirmation times, we utilized the web3.js library.

According to the obtained results, IOTA exhibits significantly higher throughput relative to Ethereum, highlighting its potential suitability for applications requiring fast transaction processing and scalability. As illustrated in the graph in Fig 11, for a number of transactions equal to 30, Ethereum achieves approximately 6.7% of the throughput compared to IOTA, which achieves around 93.3% higher throughput.

Since integrating blockchain with IoT faces challenges such as high transaction costs, low throughput, and increased energy demands, selecting the appropriate blockchain platform for IoT integration is crucial. Our evaluation indicates that IOTA outperforms Ethereum in terms of throughput and efficiency, making it a more suitable choice for applications in the harvest supply chain. Consequently, we decided to further explore the potential of IOTA's Tangle technology and seek ways to optimize it even further. In this pursuit, we focused on the tip selection algorithm, which is central to the Tangle's operation.

## 4.4 AA-WRW evaluation

The tip selection algorithm is responsible for choosing which unconfirmed transactions (known as "tips") a new transaction will approve. This process is crucial for ensuring the network's security, consistency, and scalability, as it determines

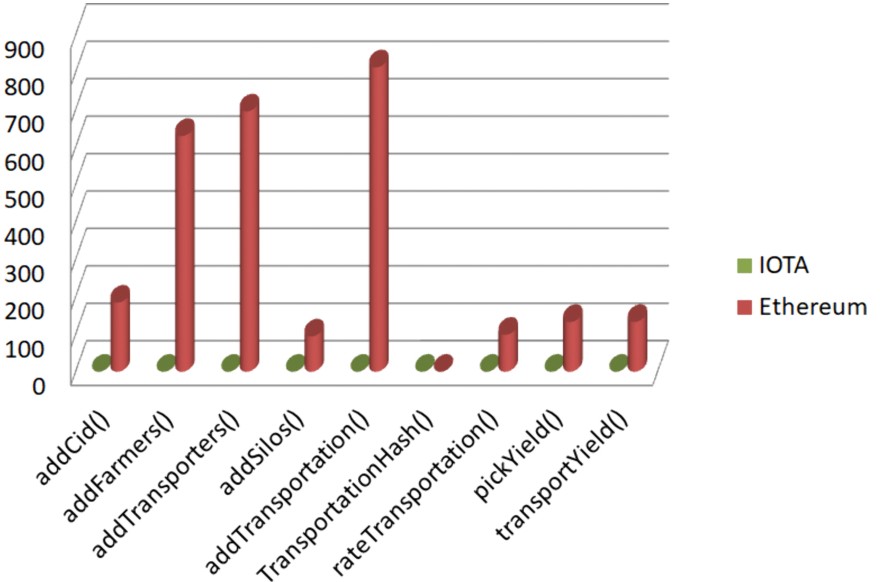

**Fig 10. Transactions costs Comparison in micro-ETH.** The figure illustrates a transaction cost comparison between the IOTA and Ethereum block-chains.The y-axis shows cost in micro-Ether (1 μETH = $10^{-6}$ ETH). Costs for IOTA functions are converted to ETH-equivalent for comparison. Ethereum transactions are much more expensive, while IOTA costs are almost negligible.

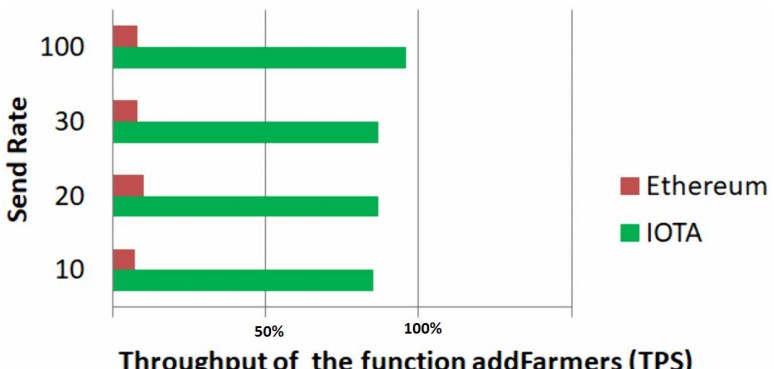

**Fig 11. Throughput Comparison for the addFarmers() function.** The figure illustrates a throughput comparison between the IOTA and Ethereum blockchains.The y-axis shows the total number of transactions sent in each test (send rate), and the x-axis shows throughput as a percentage of suc-cessfully processed transactions. Each horizontal bar represents the performance of a blockchain at a given send rate.

how quickly transactions are validated and integrated into the Tangle. Using the IOTA Tangle simulator, IOTAvisualisa-tion [35], we tested the AA-WRW algorithm within our smart contract, as demonstrated by the two functions addFarmers and addTransportations, outlined below in Figs 12 and 13, to highlight the effectiveness of this algorithm in practical use cases.

Based on the simulation results, the AA-WRW algorithm shows a significant improvement in reducing the number of permanent tips compared to the traditional WRW algorithm. Specifically, the AA-WRW algorithm achieves a reduction of approximately 24% in permanent tips for the addFarmers function and 28.03% for the addTransportation function. This reduction in permanent tips is crucial for enhancing the efficiency and scalability of the IOTA Tangle.

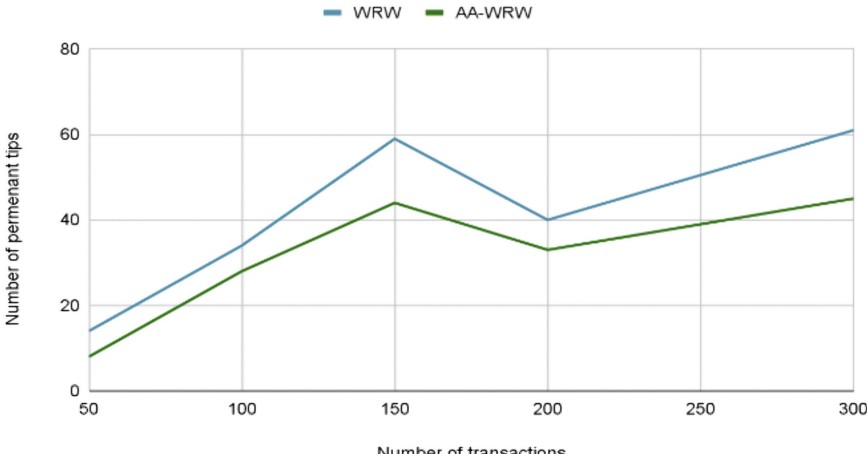

**Fig 12. Evolution of the Number of PT in the addFarmers function.** The figure compares the number of permanent tips (PT) versus the total number of transactions for two tip selection algorithms: the standard WRW and the optimized AA-WRW in the addFramers function.

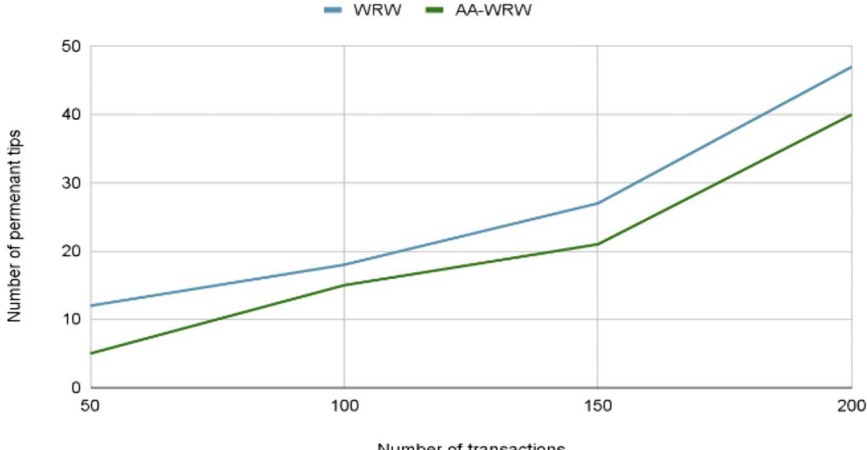

**Fig 13. Evolution of the Number of PT in the addTransportation Function.** The figure compares the number of permanent tips (PT) versus the total number of transactions for two tip selection algorithms: WRW and AA-WRW in the addTransportation function.

The AA-WRW algorithm introduces age-awareness into the tip selection process, giving higher preference to older, unconfirmed transactions with greater cumulative weight. This approach increases the likelihood of these transactions being confirmed over time, thereby minimizing the number of permanent tips left unreferenced in the network. Looking ahead, future research could focus on optimizing the age-awareness parameters, such as dynamically adjusting the weight assigned to older transactions based on network conditions. Additionally, further exploration of the algorithm's performance under varying network loads and tip selection scenarios would help ensure its robustness in real-world applications.

## 5 Conclusions

In conclusion, integrating advanced technologies such as IoT, IOTA, IPFS, AI, and SSI presents a robust solution for enhancing harvest traceability. This approach not only improves the reliability and transparency of the traceability process

but also establishes a scalable and secure framework adaptable to diverse applications within the supply chain and beyond. Specifically, our use of IOTA in the blockchain layer ensures efficient and scalable transaction management, facilitating secure and tamper-proof data exchange. Moreover, it's worth noting that IOTA continues to evolve and explore new possibilities, which may further enhance its capabilities in IoT integration.

Future enhancements in IOTA's consensus mechanisms and scalability may open up even more possibilities for its application in supply chains and other sectors reliant on secure, scalable, and feeless data transfer. As these advancements unfold, the potential for further innovation and efficiency gains will only grow, making this integrated approach increasingly valuable for businesses seeking to modernize their operations Furthermore, the integration of decentralized storage through IPFS and the use of Self-Sovereign Identity (SSI) adds additional layers of privacy and control, allowing users to maintain ownership of their data while ensuring traceability and compliance with regulatory standards. Meanwhile, AI-driven analytics can be applied to process the massive datasets generated by IoT devices, providing valuable insights that help optimize operations, predict trends, and manage risks in real time.

## Author contributions

**Conceptualization:** Mariem Turki.

**Data curation:** Imen Ahmed.

**Formal analysis:** Imen Ahmed, Bouthaina Dammak.

**Funding acquisition:** Bouthaina Dammak.

**Investigation:** Imen Ahmed, Mariem Turki.

**Methodology:** Mariem Turki, Mouna Baklouti.

**Project administration:** Mouna Baklouti.

**Resources:** Bouthaina Dammak.

**Software:** Imen Ahmed.

**Supervision:** Mouna Baklouti.

**Validation:** Mariem Turki, Mouna Baklouti, Bouthaina Dammak.

**Visualization:** Mouna Baklouti, Bouthaina Dammak.

**Writing – original draft:** Imen Ahmed, Mariem Turki.

**Writing – review & editing:** Mariem Turki, Mouna Baklouti, Bouthaina Dammak.

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
