## [Decision Letter · Decision Letter 0]

30 Sep 2025

Dear Dr. Ahmed,

plosone@plos.org. . . . A rebuttal letter that responds to each point raised by the academic editor and reviewer(s). You should upload this letter as a separate file labeled 'Response to Reviewers'.A marked-up copy of your manuscript that highlights changes made to the original version. You should upload this as a separate file labeled 'Revised Manuscript with Track Changes'.An unmarked version of your revised paper without tracked changes. You should upload this as a separate file labeled 'Manuscript'.

We look forward to receiving your revised manuscript.

Kind regards,

Sarah Jose, Ph.D.

Staff Editor

PLOS ONE

Journal Requirements:

5. We note that Figures 3-8 in your submission contain copyrighted images. All PLOS content is published under the Creative Commons Attribution License (CC BY 4.0), which means that the manuscript, images, and Supporting Information files will be freely available online, and any third party is permitted to access, download, copy, distribute, and use these materials in any way, even commercially, with proper attribution. For more information, see our copyright guidelines: http://journals.plos.org/plosone/s/licenses-and-copyright.

1. You may seek permission from the original copyright holder of Figures 3-8 to publish the content specifically under the CC BY 4.0 license.

6. We note that Figure 8 your submission contain map images which may be copyrighted. All PLOS content is published under the Creative Commons Attribution License (CC BY 4.0), which means that the manuscript, images, and Supporting Information files will be freely available online, and any third party is permitted to access, download, copy, distribute, and use these materials in any way, even commercially, with proper attribution. For these reasons, we cannot publish previously copyrighted maps or satellite images created using proprietary data, such as Google software (Google Maps, Street View, and Earth). For more information, see our copyright guidelines: http://journals.plos.org/plosone/s/licenses-and-copyright.

1. You may seek permission from the original copyright holder of Figure 8 to publish the content specifically under the CC BY 4.0 license.

Additional Editor Comments:

Reviewers' comments:

Reviewer's Responses to Questions

**Comments to the Author**

1. Is the manuscript technically sound, and do the data support the conclusions?

Reviewer #1: Yes

2. Has the statistical analysis been performed appropriately and rigorously?

Reviewer #1: N/A

3. Have the authors made all data underlying the findings in their manuscript fully available?

Reviewer #1: No

4. Is the manuscript presented in an intelligible fashion and written in standard English?

Reviewer #1: Yes

Reviewer #1: The abstract states that no previous implementation of a harvest supply chain in agriculture using IOTA smart contracts exists; the introduction mentions other blockchain-based solutions for supply chains. Therefore, the authors are required to make a clearer distinction of IOTA's unique advantages beyond general DLT benefits.

The manuscript mentions an AI model that evaluates wheat photos, but further details on its architecture (beyond CNN) and training data would be better.

Explain the specific benefits or reasons for selecting the Veramo framework over other SSI implementations.

Although a Mythril analysis was performed with no issues detected, provide a brief discussion of potential attack vectors for smart contracts in this context and how they are reduced.

Probably in Section 4.3 or a dedicated security analysis section, address cybersecurity measures and potential threats during data transactions on the blockchain.

The description of IoT sensors (DHT11, HX711, Neo-7M GPS, Pi Camera) is satisfactory, but additional specifics on their deployment, calibration, and data collection frequency may improve the section.

The paper notes that the current implementation confines each node to a single instance for testing, which should be discussed as a limitation for real-world scalability and decentralization.

IOTA's energy consumption is 13% lower than Ethereum's; a more comprehensive examination of the practical implications of this difference on large-scale agricultural operations may be necessary.

The conclusion mentions enhancing supply chain resilience and transparency in Tunisia, but a brief statement on how these findings could be generalized to other agricultural contexts or regions would broaden the paper's impact.

You may refer to the following paper on blockchain cybersecurity.

Rajababu, D. et al. (2025) “Blockchain for Cybersecurity_ Securing Data Transactions and Enhancing Privacy in Digital Systems,” in 2025 First International Conference on Advances in Computer Science, Electrical, Electronics, and Communication Technologies (CE2CT). 2025 First International Conference on Advances in Computer Science, Electrical, Electronics, and Communication Technologies (CE2CT), Bhimtal, Nainital, India: IEEE, pp. 1426–1430. Available at: https://doi.org/10.1109/CE2CT64011.2025.10939863.

.

Reviewer #1: No

---

## [Author Response · Author response to Decision Letter 1]

16 Dec 2025

Dear Reviewers,

We sincerely appreciate the time and effort you dedicated to reviewing our paper and for providing such thoughtful and constructive feedback. Your comments have been invaluable in guiding the improvements made in this revised version. The authors have carefully considered each of your suggestions and have done their best to address them thoroughly. We hope that the revised manuscript meets your expectations. We also remain open to any additional constructive feedback you may wish to share.

Below, we provide our detailed point-by-point responses. All changes in the manuscript are highlighted in red.

Best regards,

Response to Editor :

Concern # 1 : “ Please ensure that your manuscript meets PLOS ONE's style requirements, including those for file naming. “

Author response: We thank the Editor for this reminder. We have carefully reviewed the PLOS ONE style guidelines and updated the manuscript files accordingly

Concern # 2 : “ Please note that funding information should not appear in any section or other areas of your manuscript. We will only publish funding information present in the Funding Statement section of the online submission form. Please remove any funding-related text from the manuscript. “

Author response: We thank the Editor for this clarification. All funding-related text has been removed from the manuscript to comply with PLOS ONE guidelines.

Concern # 3 : “ We note that the grant information you provided in the ‘Funding Information’ and ‘Financial Disclosure’ sections do not match. “

Author response: We thank the Editor for noting this issue. The inconsistency has been corrected.

Concern # 4 : “ We note that you have indicated that there are restrictions to data sharing for this study. PLOS only allows data to be available upon request if there are legal or ethical restrictions on sharing data publicly. “

Author response: We thank the Editor for this reminder. There are no legal or ethical restrictions on sharing the anonymized dataset used in this study.

Concern # 5 : “ We note that Figures 3-8 in your submission contain copyrighted images. All PLOS content is published under the Creative Commons Attribution License (CC BY 4.0), which means that the manuscript, images, and Supporting Information files will be freely available online, and any third party is permitted to access, download, copy, distribute, and use these materials in any way, even commercially, with proper attribution. You may seek permission from the original copyright holder of Figures 3-8 to publish the content specifically under the CC BY 4.0 license… “

Author response: We thank the Editor for this observation. We confirm that Figures 3–8 in the revised manuscript are original figures created entirely by the authors and do not contain any copyrighted material. To ensure full compliance with the CC BY 4.0 license requirements, we have verified and updated all figures accordingly. Additionally, Figures 2 and 8 have been replaced in the revised submission to ensure that all visual elements are fully original and copyright-free.

Reviewer #1

« The abstract states that no previous implementation of a harvest supply chain in agriculture using IOTA smart contracts exists; the introduction mentions other blockchain-based solutions for supply chains. Therefore, the authors are required to make a clearer distinction of IOTA's unique advantages beyond general DLT benefits.

The manuscript mentions an AI model that evaluates wheat photos, but further details on its architecture (beyond CNN) and training data would be better.

Explain the specific benefits or reasons for selecting the Veramo framework over other SSI implementations.

Although a Mythril analysis was performed with no issues detected, provide a brief discussion of potential attack vectors for smart contracts in this context and how they are reduced.

Probably in Section 4.3 or a dedicated security analysis section, address cybersecurity measures and potential threats during data transactions on the blockchain.

The description of IoT sensors (DHT11, HX711, Neo-7M GPS, Pi Camera) is satisfactory, but additional specifics on their deployment, calibration, and data collection frequency may improve the section.

The paper notes that the current implementation confines each node to a single instance for testing, which should be discussed as a limitation for real-world scalability and decentralization.

IOTA's energy consumption is 13% lower than Ethereum's; a more comprehensive examination of the practical implications of this difference on large-scale agricultural operations may be necessary.

The conclusion mentions enhancing supply chain resilience and transparency in Tunisia, but a brief statement on how these findings could be generalized to other agricultural contexts or regions would broaden the paper's impact.

You may refer to the following paper on blockchain cybersecurity.

Rajababu, D. et al. (2025) “Blockchain for Cybersecurity_ Securing Data Transactions and Enhancing Privacy in Digital Systems,” in 2025 First International Conference on Advances in Computer Science, Electrical, Electronics, and Communication Technologies (CE2CT). 2025 First International Conference on Advances in Computer Science, Electrical, Electronics, and Communication Technologies (CE2CT), Bhimtal, Nainital, India: IEEE, pp. 1426–1430. Available at: https://doi.org/10.1109/CE2CT64011.2025.10939863. »

Response to Reviewer

Concern # 1

“ The abstract claims no previous implementation of a harvest supply chain using IOTA exists, but the introduction mentions other blockchain supply chains. You must clarify what makes IOTA uniquely advantageous, not just “another DLT.” “

Author response: We thank the reviewer for this insightful observation. The specific advantages of IOTA beyond general DLT benefits were already discussed in Section 2.3.2, “IOTA Tangle Suitability for IoT.” This section details IOTA’s unique characteristics, including its feeless transactions, scalability through the Directed Acyclic Graph (DAG) structure known as the Tangle, lightweight consensus mechanism adapted to IoT environments, and lower energy consumption compared to conventional blockchains such as Ethereum.

In addition, we have revised the introduction to further emphasize IOTA’s unique characteristics beyond general DLT benefits.

Concern # 2

“ Provide more details on the CNN model and the training dataset used to evaluate wheat photos. “

Author response: We thank the reviewer for this helpful comment. Although the AI component is not the main focus of the paper, its role is limited to supporting the IoT-based monitoring system, we have expanded Section 2.1 to provide additional clarity regarding both the dataset and the CNN architecture.

Concern # 3

“ Explain the benefits or reasons for selecting the Veramo framework over other SSI implementations.“

Author response: We appreciate the reviewer’s careful observation. We selected the Veramo framework due to its interoperability with W3C Decentralized Identifier (DID) standards and its modular API architecture that simplifies integration with IOTA’s distributed ledger. Compared to alternatives such as Hyperledger Indy, Veramo offers greater flexibility for lightweight IoT devices and faster DID resolution, which are critical in real-time agricultural supply chain scenarios. This explanation has been added in Section 3.3.

Concern # 4

“ Although Mythril found no issues, discuss potential attack vectors and how they’re mitigated. Add a dedicated section discussing cybersecurity measures during blockchain data transactions. “

Author response: We sincerely thank the reviewer for this valuable suggestion. We have added an explanation in Section 3.2.1 discussing potential smart contract vulnerabilities and cybersecurity measures.

Concern # 5

“ Add more specifics about how sensors were deployed, calibrated, and how frequently they collected data. “

Author response: We thank the reviewer for this valuable comment. Section 3.1 has been expanded to describe sensor deployment and data collection frequency. Each DHT11 and HX711 sensor was calibrated using manufacturer-provided reference values before deployment. Data were collected at 15-minute intervals, synchronized via the Neo-7M GPS timestamp to ensure temporal consistency across nodes.

Concern # 6

“ You mention that each node was tested as a single instance — discuss how this limits real-world scalability.“

Author response: We thank the reviewer for this insightful comment. For simplicity, the current prototype confines both Hornet and Wasp nodes to a single-instance deployment within the local network. This configuration was sufficient to validate the system’s functionality and end-to-end data flow but does not fully capture the decentralized nature and scalability potential of the IOTA network. In future work, we plan to deploy multiple distributed nodes across different devices to evaluate network throughput, latency, and resilience under realistic operational conditions.

Concern # 7

“ IOTA uses 13% less energy than Ethereum — explain why this matters in practice.“

Author response: We thank the reviewer for this valuable comment. We have expanded the discussion in Section 4.1 to clarify the practical implications of IOTA’s lower energy consumption. Specifically, we emphasize that this efficiency is particularly beneficial for large-scale agricultural and IoT-based supply chains, where thousands of daily transactions can accumulate significant energy costs. Reduced energy usage not only lowers operational expenses but also supports the deployment of lightweight, low-power IoT nodes and aligns with sustainability goals for green agriculture.

Concern # 8

“ Add a short statement on how findings can be generalized beyond Tunisia..“

Author response: We thank the reviewer for this comment. We have added a statement in the Introduction emphasizing that, although our study focuses on Tunisia, the proposed IoT and IOTA-based wheat supply chain framework can be generalized to other agricultural contexts and staple crops, where transparency, resilience, and real-time monitoring are important.

---

## [Decision Letter · Decision Letter 1]

15 Jan 2026

IOTA Tangle-based traceability framework for wheat crop supply chain

PONE-D-25-42551R1

Dear Dr. Ahmed,

We’re pleased to inform you that your manuscript has been judged scientifically suitable for publication and will be formally accepted for publication once it meets all outstanding technical requirements.

Kind regards,

Sohail Saif, Ph.D

Academic Editor

PLOS One

Additional Editor Comments (optional):

Revision is satisfactory.

Reviewers' comments:

Reviewer's Responses to Questions

**Comments to the Author**

Reviewer #1: All comments have been addressed

2. Is the manuscript technically sound, and do the data support the conclusions?

Reviewer #1: Yes

3. Has the statistical analysis been performed appropriately and rigorously?

Reviewer #1: Yes

4. Have the authors made all data underlying the findings in their manuscript fully available?

Reviewer #1: Yes

5. Is the manuscript presented in an intelligible fashion and written in standard English?

Reviewer #1: Yes

Reviewer #1: The authors have addressed all of my concerns. I suggest the authors should mention the author's name of the reference when they cite.... Like according to XXX et al., instead of according to [1]... please follow the general style of citation.

.

Reviewer #1: No
